# Ultrafast and Facile Synthesis of (Ni/Fe/Mo)OOH on Ni Foam for Oxygen Evolution Reaction in Seawater Electrolysis

Li Xu [1,2,3,4], Yuxuan Dong [1,2,3], Wei Xu [4] and Wen Zhang [1,2,3,*]

1   School of Chemical Engineering and Technology, Tianjin University, Tianjin 300350, China; xuli620@tju.edu.cn (L.X.); yuxuan_dong@tju.edu.cn (Y.D.)
2   National Industry-Education Integration Platform of Energy Storage, Tianjin University, Tianjin 300350, China
3   Tianjin Key Laboratory of Membrane Science and Desalination Technology, Tianjin 300350, China
4   Tianjin Mainland Hydrogen Equipment Co., Ltd., Tianjin 301609, China; xuwei@cnthe.cn
*   Correspondence: zhang_wen@tju.edu.cn

**Abstract:** Preparing high-performance electrocatalysts for oxygen evolution reaction (OER)s with high durability for seawater electrolysis is of great significance. Herein, reported a one-step solution-immersion synthesis strategy to prepare a (Ni/Fe/Mo)OOH catalyst on a nickel foam substrate that can be accomplished in 5 min under ambient temperature and pressure. The unique cluster morphology of the catalyst on the surface of electrodes effectively increases the number of active sites, and the presence of Mo, Ni, and Fe in the catalyst enhances the activity of the OER. In the electrolyte solution (1 mol/L NaOH), the electrode exhibited low OER overpotentials of 265 mV, 286 mV, and 332 mV at currents of 100 mA·cm$^{-2}$, 400 mA·cm$^{-2}$, and 1000 mA·cm$^{-2}$, respectively. This electrode also demonstrated excellent performance in seawater splitting, and the overpotentials at currents of 100 mA·cm$^{-2}$, 400 mA·cm$^{-2}$, and 1000 mA·cm$^{-2}$ in alkaline seawater environments were 330 mV, 416 mV, and 514 mV, respectively. In the 72 h durability test, the voltage increase was within 10 mV, exhibiting the excellent durability of the (Ni/Fe/Mo)OOH electrocatalyst. Therefore, the electrode developed here shows potential in the application of seawater electrolysis for hydrogen generation.

**Keywords:** oxygen evolution reaction; (Ni/Fe/Mo)OOH catalyst; ultrafast and facile synthesis; alkaline seawater electrolysis





## 1. Introduction

As a clean and low-carbon secondary energy source, hydrogen energy is an important carrier for building a diversified comprehensive energy supply system [1]. Hydrogen production via water electrolysis can avoid the consumption of fossil resources and emissions of CO$_2$ [2]. The generation of solar and wind power at sea has the problems of unstable power generation and difficulty in transmission and utilization. An interesting strategy for the efficient utilization of these renewable energies at sea is the in situ generation of hydrogen via seawater electrolysis [3].

Seawater electrolysis consists of two reactions, which require a large energy input [4]. Due to the rigid O-O bond and complicated processes for the transfer of protons and electrons, the reaction of the OER is relatively slow, which hinders efficient water electrolysis [5,6]. Hence, efforts should be made to develop high-performance anodic electrocatalysts to accelerate the reaction kinetics. Noble metal-based electrocatalysts, including iridium oxide (IrO$_2$) and ruthenium oxide (RuO$_2$), have good OER performance, but their commercial use is difficult due to their high cost and low total reserves in the Earth's crust [7,8]. Therefore, many efforts have been made in recent years to use nonprecious metal-based catalysts.

Ni-based catalysts are effective nonprecious metal OER electrocatalysts [9,10]. Some studies have reported that Ni-M (M = Fe, Co, Mo) oxides have superior water oxidation activity [11–13]. Research has also shown that Mo can greatly enhance the catalytic activity

of the OER because the introduction of Mo atoms can increase the electron transfer efficiency [14–16]. Overall, the preparation of Ni-based catalysts mainly focuses on designing better surface morphologies to enhance the number of exposed sites [17] and tuning the electronic structures by creating defects on the surface [18], while the integration of carbon materials improves electron transfer [19]. The OER electrode performance of these Ni-based catalysts is excellent. However, there are few studies on the time and energy costs required for the preparation of Ni-based catalysts, which is of great importance for reducing the cost of seawater electrolysis.

At present, the preparation methods of Ni-based catalysts are often complicated, requiring multiple steps, long preparation times, high temperatures, and high-pressure conditions [16]. The operation of electrodeposition is relatively simple, but complex equipment is usually involved [20]. Therefore, these methods are uneconomical for use in the large-scale preparation of Ni-based electrodes. Hence, simple and low-cost preparation methods are still needed to develop novel Ni-based electrodes for commercial applications.

Here, we used a one-step solution-immersion reaction at room temperature and atmospheric pressure to prepare cluster-like (Ni/Fe/Mo)OOH catalysts on nickel foam substrates. The preparation time was only approximately 5 min. The equipment and reagents used in the method were also easy and inexpensive to obtain. In this project, the Ni foam was used both as a substrate and a source of nickel, and the formed (Ni/Fe/Mo)OOH clusters could be firmly bonded with the substrate with excellent stability. This catalyst also demonstrated a low overpotential and good durability in a seawater environment during the long-term test.

## 2. Results and Discussion

### 2.1. Electrode Characterization

The (oxy)hydroxide of the Ni/Fe/Mo electrocatalyst was synthesized using a straightforward solution-immersion strategy, and we named the electrocatalyst (Ni/Fe/Mo)OOH. During the reaction step, the Ni foam (NF, Figure S1) acted as the source of nickel and the skeleton to grow the (Ni/Fe/Mo)OOH electrocatalysts. The color of NF changed from the initial metallic color to a dark hue from 1 to 5 min (Figure S2).

According to the SEM images in Figure 1a–c, as the reaction time increased, the etching degree of the nickel framework became obvious, and the amount of catalyst attached to the surface of the framework increased. The electrode in the 5 min group has the most uniform, dense, and neat catalyst distribution on its surface.

Figure 1d,e shows the SEM images of the electrocatalyst sample after a reaction time of 5 min. In Figure 1d, the diameter of the cluster structures is approximately 1 μm–10 μm, and in Figure 2e, the diameter of the single catalyst particle is approximately 100 nm. It can also be seen that the distribution of the catalyst is relatively uniform. The catalyst particles and clusters attached to the skeleton of the nickel foam can offer abundant active sites for the OER.

Figure 1f displays the EDS line scan image of the electrode surface. It can be clearly seen that the area without cluster aggregation is mainly composed of nickel, while in the area with cluster aggregation, the contents of molybdenum and oxygen increased significantly, and the content of iron increased slightly. Figures 1g and S3 display the elemental mapping images of (Ni/Fe/Mo)OOH. The distributions of Ni, Mo, and O are uneven. The regions with clusters are full of Mo and O, and the remaining regions without clusters are full of Ni. The distribution of Fe is relatively uniform.

We placed the prepared electrode in a beaker filled with alcohol and treated it with an ultrasonic instrument for 8 h at a power of 400 W. Fragments dropped into the bottom of the beaker. Figure 2 shows the TEM images of the fragments. In Figure 2a, the catalyst particles appear porous, with a large number of mesopores (20–50 nm). Thus, this method of surface engineering quickly transforms the even surface of NF into layers of nanoparticles with porous structures, leading to the creation of more active sites and facilitating the escape of the oxygen bubbles from the catalyst during the OER reaction. These effects prove

to be advantageous to the OER process, particularly in situations involving high current densities [21].

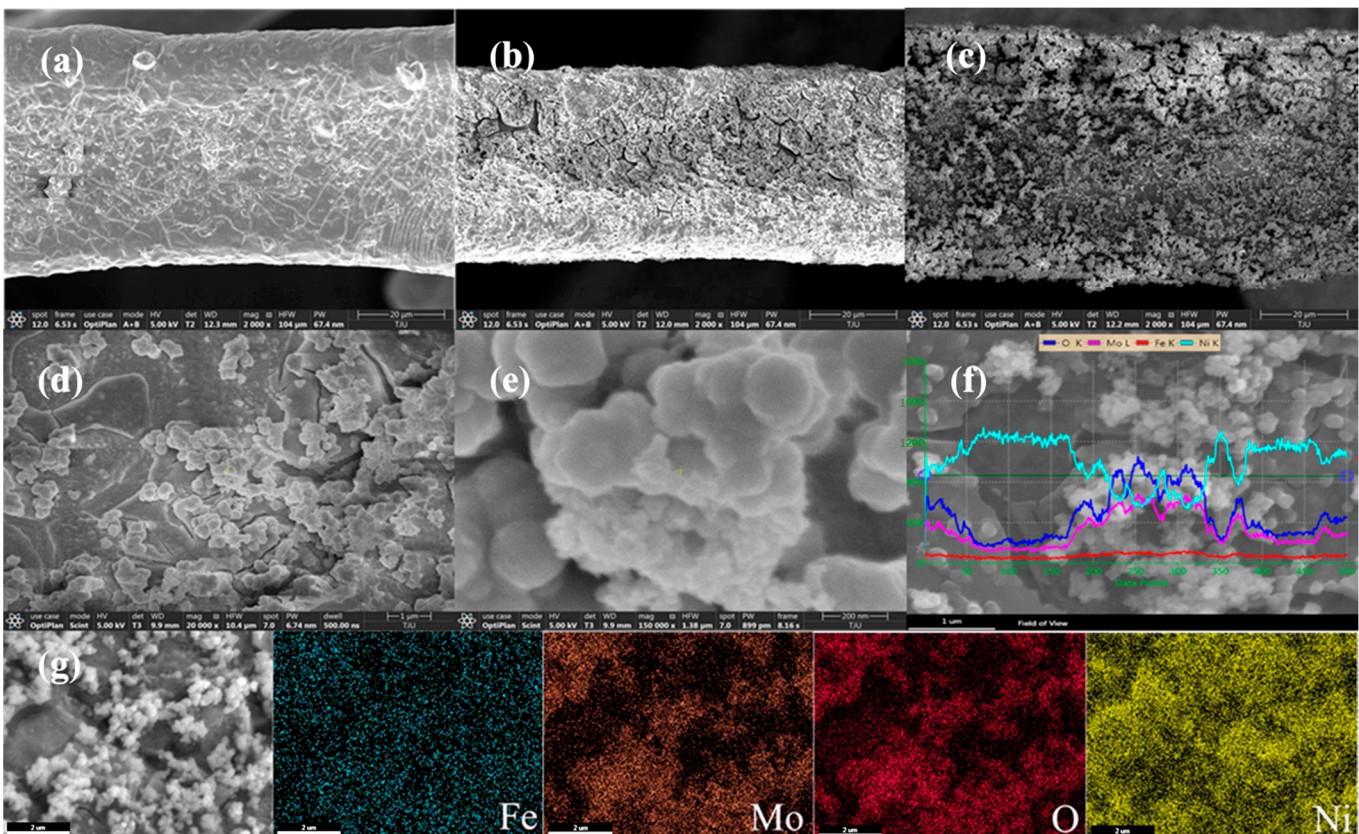

**Figure 1.** SEM image of (Ni/Fe/Mo)OOH at various immersing times: (**a**) 1 min, (**b**) 3 min, (**c**) 5 min; (**d**,**e**) SEM, (**f**) EDS line and (**g**) EDS mapping of the (Ni/Fe/Mo)OOH samples with the immersing time of 5 min.

Figure 2b shows that there are dark particles scattered in the bright area of the catalyst, and HRTEM and SAED tests were performed to study them. In Figure 2c, the dark particles show a lattice arrangement, with an interplanar distance of 0.20 nm, which corresponds to the (111) crystal face of nickel metal (PDF# 04-0850). The bright areas in the image show the amorphous morphology. In Figure 2d, the SAED analysis of the dark particles shows a diffraction ring indexed to the (111) lattice plane of nickel metal.

Figure 2e displays STEM and EDS element mapping of (Ni/Fe/Mo)OOH, and nickel, iron, and molybdenum are all distributed in the catalyst. The calculated weight ratio of nickel–molybdenum–iron was 42:60:1, and the atomic ratio was 40:35:1 (Figure S4).

We used the XRD pattern to study the structure of (Ni/Fe/Mo)OOH. As shown in Figure 3a, the peaks presented on the (Ni/Fe/Mo)OOH at 44.62° (111), 51.99° (200), and 76.49° (220) correspond to metallic Ni (PDF# 04-0850). This result is consistent with the TEM images, which show the presence of nickel metal particles in the catalyst.

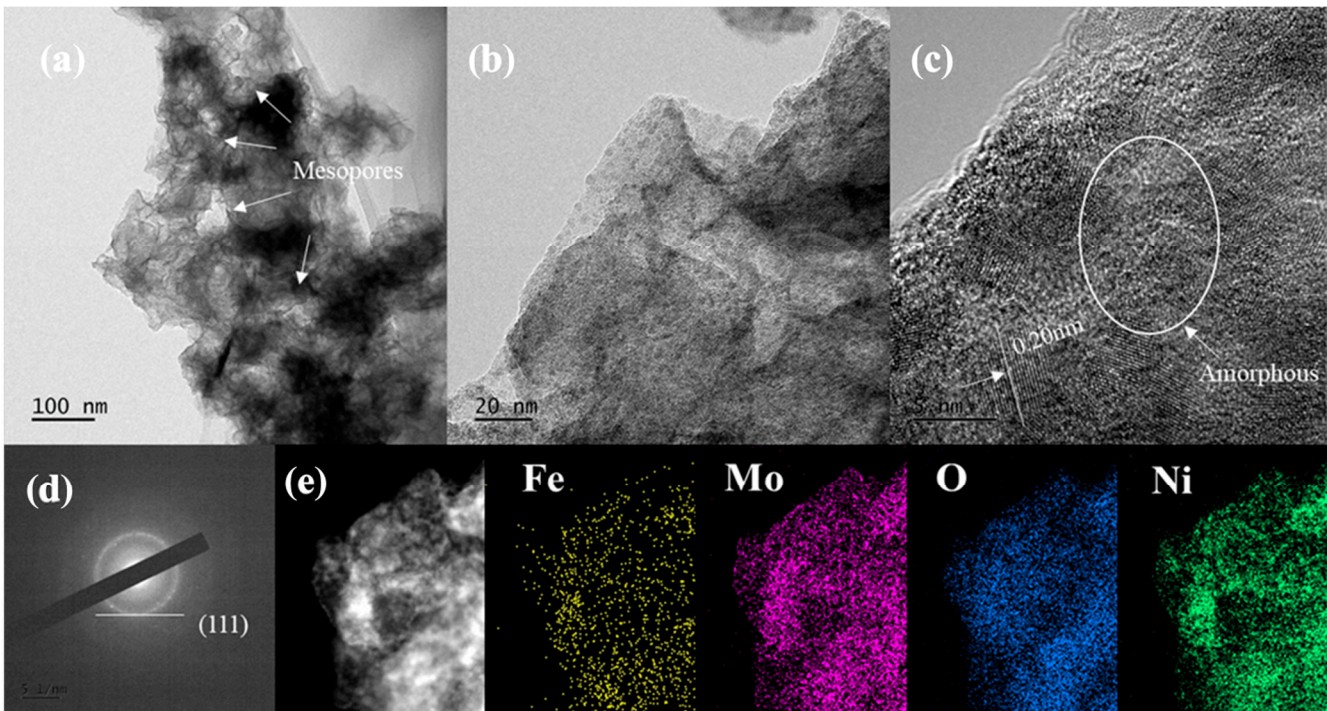

**Figure 2.** (**a**–**c**) TEM images, (**d**) selected area electron diffraction (SAED) diffraction patterns, and (**e**) STEM elemental mapping of the fragments exfoliated from the (Ni/Fe/Mo)OOH sample with the immersing time of 5 min.

We used XPS to study the valence state of elements in (Ni/Fe/Mo)OOH. In Figure 3b, XPS verified the existence of elemental Ni, Fe, Mo, and O. Figure 3c displays the main BE of Ni $2p_{1/2}$ and Ni $2p_{3/2}$ at 873.6 eV and 856.2 eV, respectively, suggesting the presence of $Ni^{2+}$ ions [22]. The peaks at 861.7 eV and 880.0 eV could be interpreted as shakeup satellite peaks [23]. The peaks at 853.2 eV and 870.5 eV can be interpreted as $Ni^0$, indicating the presence of metallic Ni [22]. From the Ni XPS spectrum, most of the Ni species in the catalyst were $Ni^{2+}$, and there was only a small fraction of $Ni^0$ species. According to the TEM and XRD results, the crystalline portion in the catalyst is Ni metal, which is the $Ni^0$ species. Hence, the $Ni^{2+}$ species exist in the amorphous regions of the catalyst. In Figure 3d, the peaks at 713.8 eV and 726.3 eV correspond to $Fe^{3+}$ $2p_{3/2}$ and $Fe^{3+}$ $2p_{1/2}$, respectively [24]. In Figure 3e, the peaks at 232.4 eV and 235.6 eV represent $Mo^{6+}$ $3d_{5/2}$ and $Mo^{6+}$ $3d_{3/2}$, respectively [25]. That is, there are trivalent iron and hexavalent molybdenum ions in the amorphous regions of the catalyst. In Figure 3f, the peaks at 530.8 eV and 532.1 eV could be attributed to metal-O and metal-OH, respectively [26].

The $Fe(NO_3)_3$, is a strong oxidizing agent and can react with nickel quickly ($Ni + 2Fe^{3+} \rightarrow Ni^{2+} + 2Fe^{2+}$). $Ni^{2+}$ and $Fe^{2+}$ will combine with $OH^-$, and at the same time react with dissolved oxygen to form $Ni(OH)_2$ and FeOOH [27]. Molybdate ($Mo_7O_{24}^{6-}$) dissolved in water will also be oxidized by $Fe(NO_3)_3$ to produce $MoO_3$ [28]. and both of these compounds have good electrocatalytic properties [29–31]. Therefore, the composition of the electrode prepared by the one-step solution-immersion method can be inferred and includes amorphous $Ni(OH)_2$, FeOOH, $MoO_3$, and crystal Ni nanoparticles.

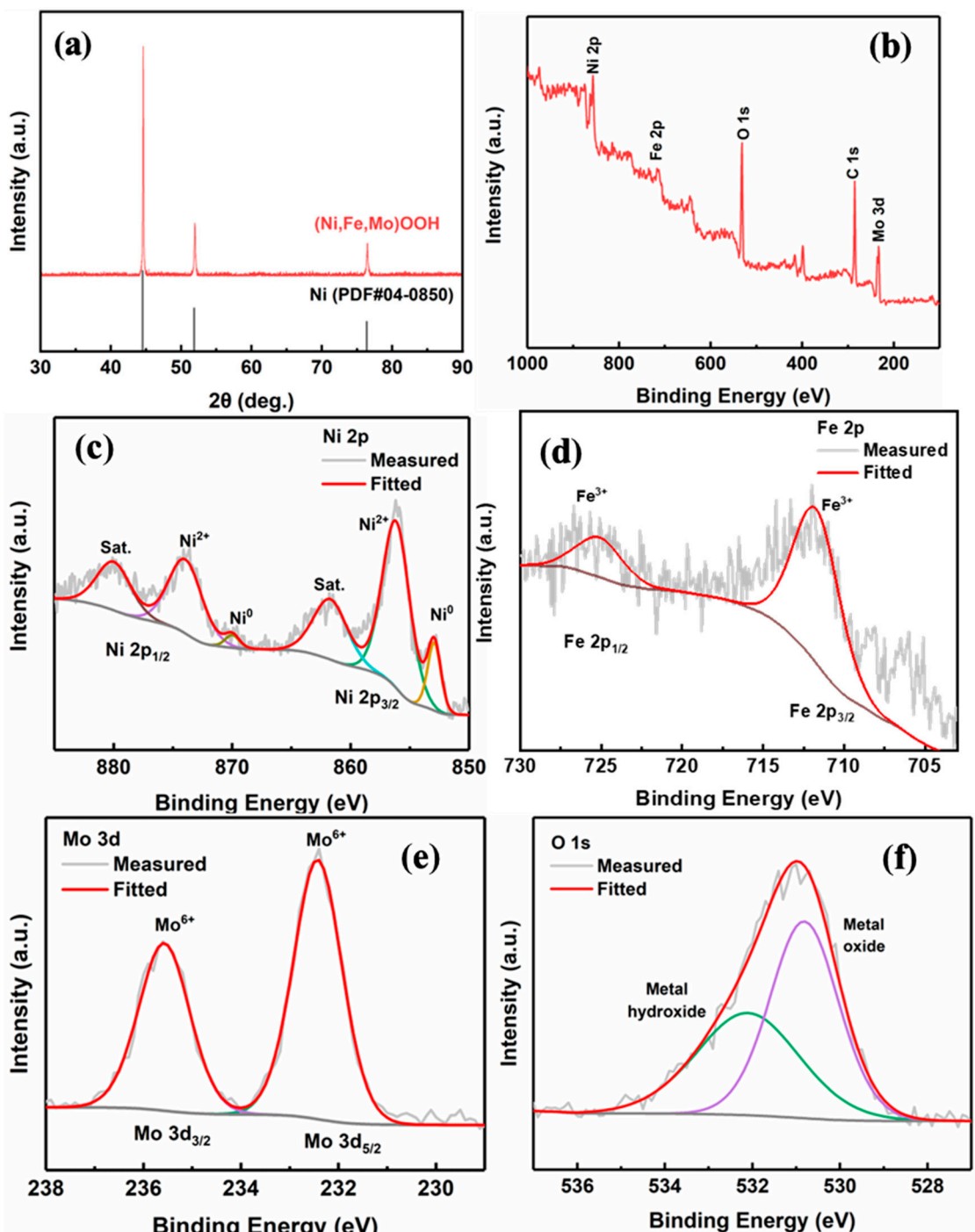

**Figure 3.** (**a**) XRD patterns, XPS of (Ni/Fe/Mo)OOH (5 min): (**b**) Survey, (**c**) Ni 2p, (**d**) Fe 2p, (**e**) Mo 3d, (**f**) O 1s.

This method is conducted at room temperature and is ultrafast and simple, without any complicated procedures. In addition, this method is straightforward to scale up, indicating its significant potential for large-scale applications in seawater electrolysis.

### 2.2. Electrocatalytic Activity for the OER

We initially tested the OER performance in a freshwater electrolyte solution of 1 mol/L KOH. As a benchmark for comparison, commercial $IrO_2$ powder loaded on NF was also tested using the same conditions. As shown in the OER polarization curve of Figure 4a,

the (Ni/Fe/Mo)OOH electrode exhibits much better performance than the conventional Ni foam electrode and the benchmark $IrO_2$ electrode. The overpotential required for the (Ni/Fe/Mo)OOH electrode to deliver a current density of 100 mA·cm$^{-2}$ is 265 mV, which is significantly lower than those required for Ni foam (450 mV) and $IrO_2$ (369 mV). Even at high currents of 400 mA/cm$^2$ and 1000 mA/cm$^2$, the overpotentials of the (Ni/Fe/Mo)OOH electrode were only 286 and 332 mV, respectively. The (Ni/Fe/Mo)OOH electrode displayed a Tafel slope of 70.4 mV·dec$^{-1}$ (Figure 4b), which is smaller than those of NF (107.1 mV·dec$^{-1}$) and $IrO_2$ (88.1 mV·dec$^{-1}$). The abovementioned result indicates the faster OER catalytic kinetics of (Ni/Fe/Mo)OOH.

In Figure 4c, the OER performance of the (Ni/Fe/Mo)OOH samples was measured in a 1 mol/L KOH electrolyte solution. As the reaction time increased, the OER performance initially improved.

We obtained double-layer capacitances ($C_{dl}$) from CV curves (Figures S5–S8). Then, we calculated the electrochemically active surface area (ECSA). In Figure 4d, with increasing etching time, the $C_{dl}$ value increased significantly. The $C_{dl}$ values for reaction times of 1, 3, and 5 min are 3.25 times, 4.98 times, and 6.42 times that of the Ni foam electrode, respectively.

Figure 4e displays the Nyquist plots obtained from electrochemical impedance spectroscopy (EIS), which demonstrate that the (Ni/Fe/Mo)OOH samples exhibit significantly smaller charge transfer resistances ($R_{ct}$) compared to the Ni foam. The $R_{ct}$ of the 5 min sample was the smallest, only 0.7 $\Omega$, which is a great improvement compared to the 13.7 $\Omega$ of nickel foam, displaying its effective charge transport ability and advantageous OER kinetics [32].

We also generated the (Ni/Fe)OOH catalyst for comparison, which was prepared using the same method without molybdenum in a reaction time of 1–5 min. As shown in the SEM image in Figure S9, the surface differences of the (Ni/Fe)OOH electrode with different reaction times were less, and there are more wrinkles but no cracks or clustered catalysts compared to the (Ni/Fe/Mo)OOH surface. Figure 4f shows that the performance of the three groups of (Ni/Fe)OOH electrodes is very close, and the 1 min group was slightly better. The results indicate that the catalysts produced have reached a stable state when the preparation time exceeds 1 min. Figure S10 shows that the (Ni/Fe)OOH electrode surface is smoother. Figure S11 proves that nickel and iron are uniformly distributed. These experiments showed that it only takes a short time during the preparation of (Ni/Fe)OOH for the surface of the catalyst to be covered with nickel skeletons and tend to be stable. Compared with the three-dimensional shape of the clustered (Ni/Fe/Mo)OOH catalyst, the smaller ECSA of (Ni/Fe)OOH will also affect the performance.

From Figure S12, it can be concluded that the composition of (Ni/Fe)OOH is very similar to that of (Ni/Fe/Mo)OOH and includes FeOOH and $Ni(OH)_2$, named as (Ni/Fe)OOH. The best-performing group of (Ni/Fe)OOH catalysts showed an overpotential of 310 mV at a current density of 100 mA·cm$^{-2}$. The overpotential of (Ni/Fe/Mo)OOH under the same test conditions was 265 mV, much better than that of the (Ni/Fe)OOH samples (Figure S13). It can be proven that the introduction of Mo into the NiFe can enhance the electrocatalysis performance. The modification with Mo and Fe has a synergistic effect that enhances both the activity and stability of the system, resulting in lower adsorption energies for OER intermediates and higher activity in the OER [33].

We also compared the (Ni/Fe/Mo)OOH catalyst with other similar catalysts. As shown in Table 1, compared with traditional electrodes, the (Ni/Fe/Mo)OOH catalyst reported here can be prepared using a cost-effective one-step strategy to grow Ni/Fe/Mo (oxy)hydroxide with porous structures in a few minutes at room temperature. Compared to the hydrothermal method, this method significantly shortens the preparation time and does not require a high-temperature and high-pressure environment. Compared to the electrodeposition method, it can eliminate the need for complicated equipment.

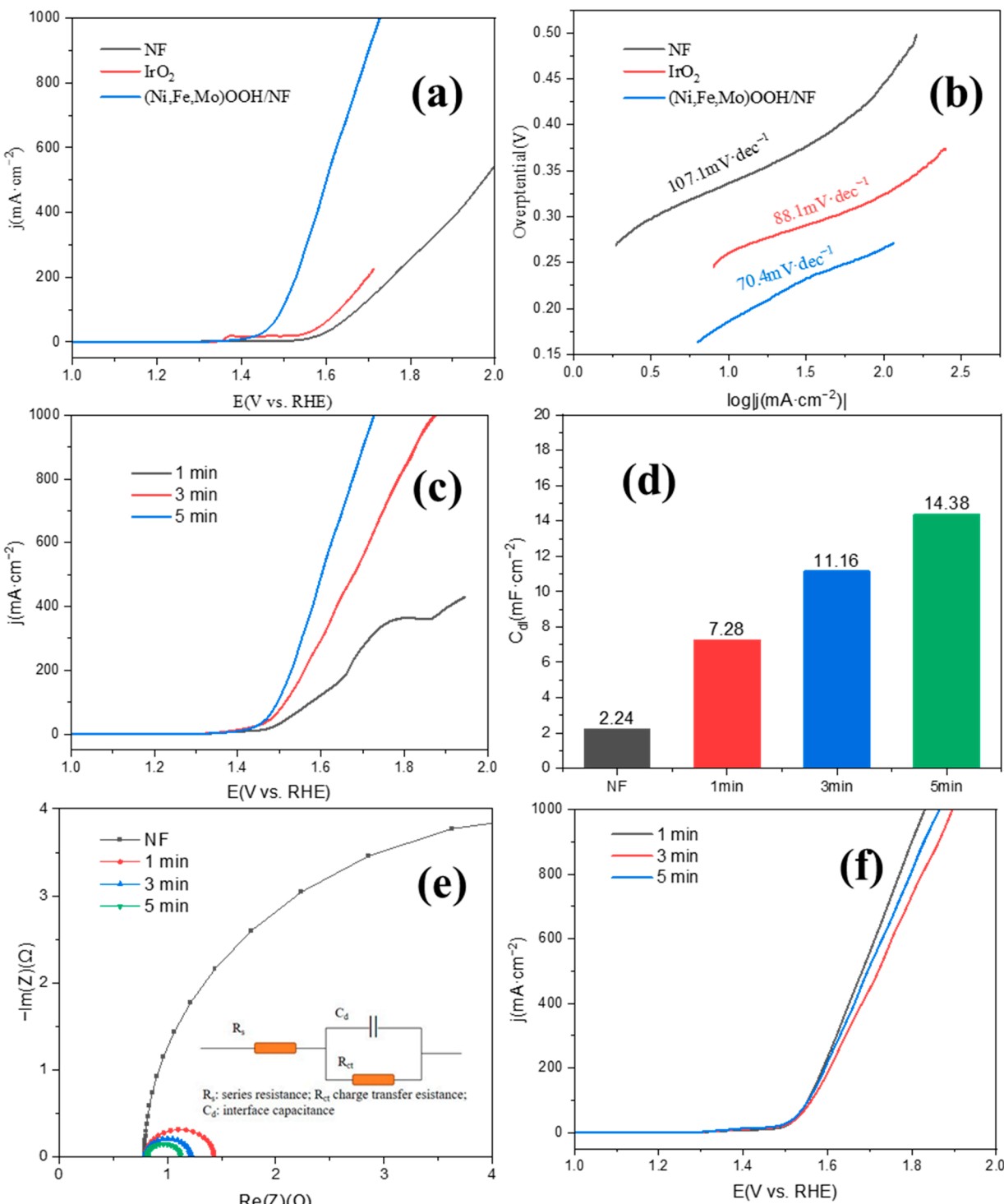

**Figure 4.** (**a**) Polarization and (**b**) Tafel curves of nickel foam, IrO$_2$, and (Ni/Fe/Mo)OOH (5 min) samples. (**c**) Polarization, (**d**) $C_{dl}$, and (**e**) Nyquist plots of the (Ni/Fe/Mo)OOH samples. (**f**) Polarization plots of (Ni/Fe)OOH samples prepared with different reaction times.

**Table 1.** Comparison of the electrocatalytic performance of (Ni\Fe\Mo)OOH with other recently reported OER catalysts.

| Electrocatalysts | Synthesis Process | Substrate | Current Density | Electrocatalysts | Classification of Synthetic Methods |
|---|---|---|---|---|---|
| (Ni/Fe/Mo)OOH | One-step solution-immersion synthesis, 5 min at room temperature. | Ni foam | 100 | 265 | One-step solution-immersion synthesis |
| S-(Ni,Fe)OOH [27] | One-pot solution-phase method: 5 min at room temperature. | Ni foam | 100 | 281 | |
| Ni/NiFeMoO$_x$ [28] | Three steps: hydrothermal for 10 h at 120 °C, 10 h at 120 °C using different solution and 2 h at 400 °C | Ni foam | 100 | 289 | Hydrothermal method |
| NiFeOH [34] | Two steps: hydrothermal for 10 h at 120 °C and 12 h at 50 °C. | Glassy carbon | 100 | 350 | |
| Se-doped FeOOH [35] | Two steps: hydrothermal for 12 h at 140 °C and 4 h at room temperature. | Fe foam | 100 | 279 | |
| FeCoNiOOH [36] | Hydrothermal for 6 h at 190 °C. | Ni foam | 100 | 330 | |
| Fe$_x$Co$_{1-x}$OOH [37] | A two-step electrochemical method: 1 h and 30 min at room temperature. | Carbon fiber cloth | 100 | 300 | Electrodeposition method |
| Ni–Fe–Mo [20] | Electrodeposition for 2 min. | Ni foam | 10 | 306 | |
| NiFeO$_x$/NiFeOOH [38] | Two steps: 1 h for acid corrosion and 10 h of electrodeposition at room temperature. | Stainless steel | 100 | 280 | |
| NiCo-LDH@NiCoV-LDH [12] | Hydrothermal for 12 h at 110 °C, then electrodeposition for 15 min. | Ni foam | 100 | 260 | |
| (Ni,Fe)OOH [39] | Two steps, including a strong mechanically stirred process: 18 h at room temperature. | Ni foam | 100 | 220 | Multi-step multi-method mixed-use |
| NiCoP [40] | Four steps: using microwave refluxing system 400 w for 20 min, hydrothermal for 10 h at 90 °C, 4 h at 400 °C, and using the expensive reagent AgNO$_3$. | Ni foam | 100 | 345 | |
| Ni:FeOOH [41] | Using hydrothermal process for 24 h at 180 °C, then using electrodeposition for 1 h at room temperature. | N-doped graphite foam | 100 | 270 | |

*2.3. Durability Testing and Seawater Splitting Performance*

Subsequently, we assessed the electrocatalytic activity of the (Ni/Fe/Mo)OOH electrode in different electrolytes. In Figure 5a,b, the overpotentials determined in a simulated high-salinity electrolyte (1 mol/L KOH + 1 mol/L NaCl) at current densities of 100 mA·cm$^{-2}$, 400 mA·cm$^{-2}$, and 1000 mA·cm$^{-2}$ were 280 mV, 315 mV, and 384 mV, respectively.

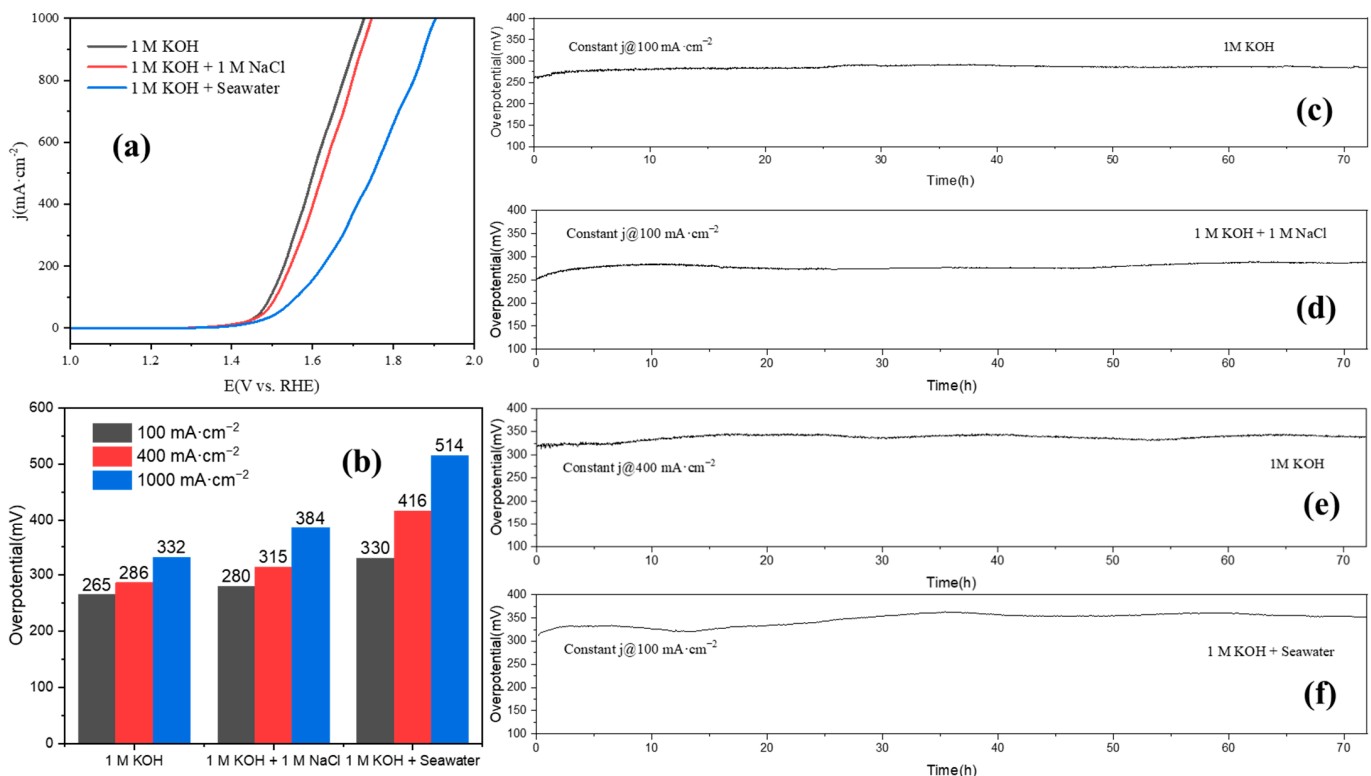

**Figure 5.** (**a**) Polarization plots and (**b**) overpotentials required to obtain current densities of 100, 400, and 1000 mA·cm$^{-2}$ for (Ni/Fe/Mo)OOH (5 min) tested in different electrolytes. Stability tests at a current density of 100 mA·cm$^{-2}$ in (**c**) 1 mol/L KOH, (**d**) 1 mol/L KOH + 1 mol/L NaCl, (**f**) 1 mol/L KOH + Seawater, and (**e**) constant current density of 400 mA·cm$^{-2}$ in 1 mol/L KOH.

In seawater containing chloride ions, the OER and CIER are competitive reactions. The standard potential of CIER is 1.36 V vs. RHE, higher than that of OER (1.23 V vs. RHE). To reduce the risk of CIER, a lower voltage (overpotential) should be maintained [42].

In the commercial application of electrolysis for hydrogen production, a current density of 400 mA·cm$^{-2}$ is commonly used. At 400 mA·cm$^{-2}$, this (Ni/Fe/Mo)OOH electrode exhibits an overpotential of 416 mV in alkaline seawater (1 mol/L KOH + seawater), which is still lower than the overpotential of 490 mV [43] that would cause a significant CIER. Therefore, this electrode has the potential for application in the field of seawater electrolysis.

The stability of the electrode was tested under different current conditions and in different solution environments for 72 h. Figure 5c shows the durability test result in 1 mol/L KOH electrolyte at a 100 mA·cm$^{-2}$ current density, and it can be seen that the voltage had almost no increase. Figures S14 and S15 are the SEM images and EDS mapping images after the test. It can be seen that the nickel skeleton still maintains its original form after the durability test. According to the XPS results in Figure S16, the composition is the same as that of the original electrodes in Figure 3. Hence, it can be judged that the electrode has excellent durability in alkaline electrolytes, and the surface morphology and composition of the electrode have not changed obviously.

Figure 5d shows the durability test result in a chlorine-containing alkaline (1 mol/L KOH + 1 mol/L NaCl) at 100 mA·cm$^{-2}$, and it can be seen that the voltage also had almost no visible increase. It can also be seen that at low current densities, this catalyst had good durability. In Figure 5e, at 400 mA·cm$^{-2}$ under 1 mol/L KOH, the voltage also had only a small variation. Figure 5f shows that in alkaline simulated seawater (1 mol/L KOH + seawater) at 100 mA·cm$^{-2}$, the voltage increase was within 10 mV, demonstrating excellent OER durability and stability. Figure S17 shows the SEM image of the (Ni/Fe/Mo)OOH electrode after 72 h of operation at 100 mA·cm$^{-2}$ in alkaline simulated seawater (1 mol/L KOH + seawater). Compared with Figure 2c–e, the surface of the electrode has many newly

formed pitted structures, which exposed the inside of the catalyst-free nickel foam skeleton and reduce the effective loading capacity of the catalyst. The newly formed small particles on the surface were slightly different from the original cluster catalysts. From the EDS image in Figure S18, it can be seen that these small particles contain a lot of calcium and magnesium, which may be due to the precipitation of hydroxides formed by the calcium and magnesium ions rich in seawater in an alkaline environment, which will destroy the original surface structure and reduce the number of active sites. This may be due to the precipitation of hydroxides formed by the abundant calcium and magnesium ions in seawater in alkaline environments, which will destroy the original surface structure and reduce the number of active sites. Currently, the use of ion exchange technology to soften water is very mature [44]. After removing the influence of alkaline earth metal ions, this electrode shows great benefits to the performance of chloride ions, and there are still good application scenarios for the electrolysis of seawater.

The electrochemical results indicated the (Ni/Fe/Mo)OOH catalyst has excellent catalytic performance and good durability for the OER in seawater and is suitable under high current densities. The excellent performance is due to the following factors: (1) the cluster-shaped (Ni/Fe/Mo)OOH catalyst provides a larger surface area and more active sites for the catalytic reaction; (2) the relatively simple surface morphology helps to facilitate efficient electrolyte flow and diffusion, contributing to improved performance at high current densities; and (3) the strong binding force between the (Ni/Fe/Mo)OOH catalyst prepared by the in situ growth method and the nickel foam can reduce the internal resistance ($R_{ct}$) can promote an improvement in electrocatalytic stability.

### 3. Experimental Sections

#### 3.1. Chemicals

Nickel foam (thickness: 1.5 mm, porosity: ~95%, Li Yuan, Changsha, China), $Fe(NO_3)_3 \cdot 9H_2O$ (AR, Aladdin, Shanghai, China), Ammonium molybdate tetrahydrate ($H_{24}Mo_7N_6O_{24} \cdot 4H_2O$, AR, Aladdin, Shanghai, China), Hydrochloric acid (HCl, AR, Damao, Tianjin, China), NaOH (97%, Rhawn, Shanghai, China), KOH (95%, Aladdin, Shanghai, China), Ethanol ($CH_3CH_2OH$, AR, Yuanli Chemical, Tianjin, China), and Deionized water (Conductivity: 1.5 $\mu S \cdot cm^{-1}$, Yuanli Chemical, Tianjin China) were used for preparing the solution.

#### 3.2. Synthesis of (Ni/Fe/Mo)OOH and (Ni/Fe)OOH Catalysts on Nickel Foam

The nickel foams were pretreated before use. They were soaked in 2 M NaOH for 1 h, rinse with DI water, and then soaked in 3 M HCL for 15 min and cleaned with water and ethanol to remove the oil and oxide film. Then, as shown in Figure 6, we used a one-step solution-immersion method at room temperature to synthesize (Ni/Fe/Mo)OOH catalyst on nickel foam. The solution was prepared by dissolving 0.25 g $H_{24}Mo_7N_6O_{24} \cdot 4H_2O$ and 1.6 g $Fe(NO_3)_3 \cdot 9H_2O$ into 40 mL DI water in a backer and then the Ni foam (10 mm × 15 mm) was put into the solutions for different times. Finally, the nickel foam was cleaned again with DI and ethanol and dried naturally in the air. For the (Ni/Fe)OOH catalyst as a control sample, the preparation method is the same as that abovementioned except that there was no $H_{24}Mo_7N_6O_{24} \cdot 4H_2O$ used in the main step.

#### 3.3. Synthesis of IrO$_2$ Electrocatalyst on Nickel Foam

First, 5 mg $IrO_2$ and 80 $\mu$L Nafion were added to the mixture of ethanol/water (750 $\mu$L/250 $\mu$L) solution in a tube, and the mixture was then ultrasonicated for half an hour. Two drops of the mixture are then applied to the nickel foam (10 mm × 15 mm) using a glue head dropper and left to dry for two minutes. We repeated this step ten times and let dry in air overnight at the end. After weighing the load of $IrO_2$ is 2.0 $mg \cdot cm^{-2}$ [45].

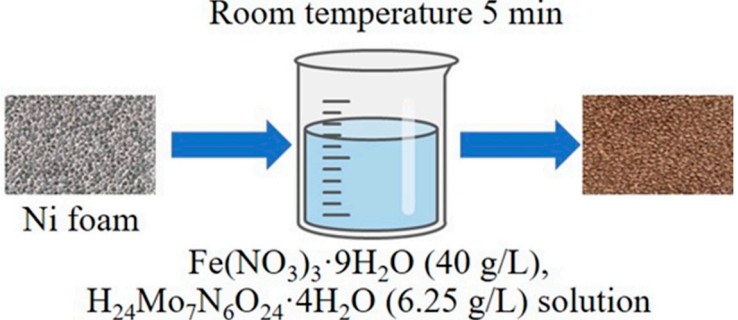

**Figure 6.** Schematic of synthetic processes of (Ni/Fe/Mo)OOH.

### 3.4. Materials Characterization

The morphology of electrocatalysts was observed by scanning electron microscopies (SEM, Apero S LoVac, FEI, Prague, Czech) and transmission electron microscopies (TEM, JEM-F200, JEOL, Tokyo, Japan). The above devices are coupled with energy dispersive spectroscopies (EDS). The structure of electrodes was characterized by the X-ray diffractometer (XRD, D8-focus, Cu Kα, λ = 0.154178 nm, Bruker AXS, Salbuluken, Germany). X-ray photoelectron spectroscopy was employed to detect the valence states of the elements (XPS, ESCALAB Xi+, Al Kα, Thermo SCIENTIFIC, Waltham, MA, USA).

### 3.5. Electrochemical Measurements

We conducted the electrochemical test at ambient temperature using an electrochemical station (VersaSTAT3, AMETEK, San Diego, CA, USA). The standard three-electrode system was employed to test the linear sweep voltammetry (LSV), Cyclic voltammetry (CV), and electrochemical impedance spectroscopy (EIS). The effective area of the prepared samples is 1 cm × 1 cm. The standard Hg/HgO electrode was the reference electrode. The graphite rod was the counter electrode. We used three electrolytes, 1 mol/L KOH, 1 mol/L KOH + 1 mol/L NaCl (pH = 14), and 1 mol/L KOH + Seawater (pH = 14). The real seawater was from the Yellow Sea near Qingdao, China. The polarization plot was measured using the scan rate of 5 mV·s$^{-1}$. The stability test was carried out under 100 and 400 mA·cm$^{-2}$. The double-layer capacitance ($C_{dl}$) value and CV curve were tested using scan rates from 40 mV·s$^{-1}$ to 90 mV·s$^{-1}$ with the interval of 10 mV·s$^{-1}$. Electrochemical impedance spectra (EIS) were obtained at an overpotential of 300 mV from 0.05 Hz to 100 kHz with an amplitude of 5 mV. The potentials vs. Hg/HgO were converted to the reversible hydrogen electrode (RHE) through the Nernst equation.

$$E_{RHE} = E_{Hg/HgO} + 0.0591 \times pH + 0.098 \tag{1}$$

The overpotential is corrected by 80% $iR$ correction according to the following equation.

$$E = E_{RHE} - 0.8 \times iR_s \tag{2}$$

where $R_s$ is the solution resistance, and $i$ is the current density [46,47].

### 4. Conclusions

In summary, this work provides an ultrafast and simple method to prepare a robust (Ni/Fe/Mo)OOH catalyst on a Ni foam substrate for efficient seawater electrolysis. The unique cluster morphology of the electrode increases the number of active sites, and Mo atoms can effectively promote electron transfer to enhance performance. In an alkaline electrolyte solution, the overpotentials for the OER at 100 mA·cm$^{-2}$ and 400 mA·cm$^{-2}$ were only 265 mV and 286 mV, respectively, and the electrode had a small charge transfer resistance ($R_{ct}$) of only 0.7 Ω. The excellent electrocatalytic property in alkaline simulated seawater (1 mol/L KOH + seawater) electrolysis and durability tests was also demonstrated, with OER overpotentials of 330 mV and 416 mV at 100 and 400 mA·cm$^{-2}$, respectively.

In the durability test at 100 mA/cm$^2$ for 72 h, the voltage increase was within 10 mV. The OER electrode used here has low fabrication costs and energy consumption for OER electrochemical performance and shows application prospects in hydrogen economy and seawater electrolysis.

**Supplementary Materials:** The following supporting information can be downloaded at: https://www.mdpi.com/article/10.3390/catal13060924/s1, Figure S1. SEM images of pre-treated Ni foam; Figure S2. Photograph of commercial Ni foam and (Ni/Fe/Mo)OOH electrodes prepared under different reaction times; Figure S3. The EDS spectrum of (Ni/Fe/Mo)OOH using SEM; Figure S4. The EDS spectrum of (Ni/Fe/Mo)OOH using TEM; Figure S5. CV curves of Ni foam, calculation of Cdl involves plotting capacitive current density against scan rate and fitting a linear regression to the resulting plot; Figure S6. CV curves of (Ni/Fe/Mo)OOH (reaction time: 1 min), calculation of Cdl involves plotting capacitive current density against scan rate and fitting a linear regression to the resulting plot; Figure S7. CV curves of (Ni/Fe/Mo)OOH (reaction time: 3 min), calculation of Cdl involves plotting capacitive current density against scan rate and fitting a linear regression to the resulting plot; Figure S8. CV curves of (Ni/Fe/Mo)OOH (reaction time: 5 min), calculation of Cdl involves plotting capacitive current density against scan rate and fitting a linear regression to the resulting plot; Figure S9. SEM images of (Ni/Fe)OOH at different immersing time, (a) 1 min, (b) 3 min, (c) 5 min; Figure S10. HR-SEM images of (Ni/Fe)OOH (5 min); Figure S11. EDS mapping images of the (Ni/Fe)OOH sample with the immersing time of 5 min; Figure S12. XPS spectra for (Ni/Fe)OOH (5 min): (a) Survey, (b) Ni 2p, (c) Fe 2p, (d) O 1s; Figure S13. Polarization curves of Ni foam, IrO2, (Ni/Fe/Mo)OOH and (Ni/Fe)OOH electrodes; Figure S14. SEM image of (Ni/Fe/Mo)OOH catalyst after 72 h of hydrogen production at a current density of 100 mA·cm$^{-2}$ in 1 M KOH electrolyte; Figure S15. EDS mapping images of the (Ni/Fe/Mo)OOH sample after 72 h of hydrogen production at a current density of 100 mA·cm$^{-2}$ in 1 M KOH electrolyte; Figure S16. XPS spectra for (Ni/Fe/Mo)OOH after 72 h of hydrogen production at a current density of 100 mA·cm$^{-2}$ in 1 M KOH electrolyte: (a) Survey, (b) Ni 2p, (c) Fe 2p, (d) Mo 3d, (e) O 1s; Figure S17. SEM image of (Ni,Fe,Mo)OOH catalyst after 72 h of hydrogen production at a current density of 100 mA·cm$^{-2}$ in alkaline natural seawater electrolyte; Figure S18. EDS mapping images of the (Ni/Fe/Mo)OOH sample after 72 h of hydrogen production at a current density of 100 mA·cm$^{-2}$ in alkaline natural seawater electrolyte.

**Author Contributions:** Conceptualization, L.X. and Y.D.; methodology, L.X. and Y.D.; software, Y.D.; validation, L.X., Y.D. and W.X.; formal analysis, Y.D.; investigation, L.X. and Y.D.; resources, L.X. and W.X.; data curation, Y.D.; writing—original draft preparation, Y.D.; writing—review and editing, W.Z.; visualization, Y.D.; supervision, L.X. and W.X.; project administration, L.X.; funding acquisition, L.X. and Y.D. All authors have read and agreed to the published version of the manuscript.

**Funding:** This research was funded by National Key R&D Program, grant number 2021YFB4000303.

**Data Availability Statement:** The data presented in this study are available on request from the corresponding author. The data are not publicly available due to further research is being done.

**Acknowledgments:** The authors thank the National Key R&D Program (No. 2021YFB4000303) for funding.

**Conflicts of Interest:** The authors declare no conflict of interest.

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
