# Peer review of "Ultrafast and Facile Synthesis of (Ni/Fe/Mo)OOH on Ni Foam for Oxygen Evolution Reaction in Seawater Electrolysis"

_catalysts, doi:10.3390/catal13060924_

Round 1

Reviewer 1 Report

The article reports one-step solution-immersion synthesis approach to obtain (Ni/Fe/Mo)OOH loaded on nickel foam, which then applied for oxygen evolution reaction in seawater electrolysis. I recommend the article if author can address the below comments.

1.      What is driving force for the growth of (Ni/Fe/Mo)OOH on Ni foam in just few minutes at room temperature.

2.     Is it possible to extend this spontaneous synthesis of (Ni/Fe/Mo)OOH electrocatalysts to grow other metal chalcogenides on NF, if it is not a selective synthesis approach.

3.     What is the active metal species in (Ni/Fe/Mo)OOH electrocatalysts for OER and its oxidation state before and after the OER study.

4.     The changes in the oxidation state before and after the OER study will be much helpful using XPS analysis.

5.     XRD peak identification for the Fe and Mo compounds is missing.

6.     The amount of active material loaded on the NF during the synthesis should be discussed.

7.     OER results of (Ni/Mo)OOH and  (Ni/Fe)OOH for comparison can be productive if it can be formed.

None

Author Response

Dear Editor:

We appreciate the opportunity to modify our paper according to the comments of the reviewer. We would like to submit the revised manuscript, along with our response to each comment made by the reviewer as shown below. All the revisions in the revised manuscript are highlighted.

We look forward to receiving your further comments on our revised manuscript. Thanks a lot.

Yours sincerely,

W. Zhang

Tianjin University

Responses to the reviewers’ comments:

Reviewer #1:

Comment 1: What is driving force for the growth of (Ni/Fe/Mo)OOH on Ni foam in just few minutes at room temperature.

Response: Thanks for you remarks. The Fe(NO3)3 has a strong oxidation property and can react with nickel quickly ().  Ni2+ and Fe2+ will combine with OH-, and react with dissolved oxygen to form Ni(OH)2 and FeOOH (Energy Environ. Sci., 2020,13, 3439-3446). Molybdate (Mo7O246-) dissolved in water will also be oxidized by Fe(NO3)3 to produce MoO3 (Adv. Sci. 2020, 7, 1902034). In this study, (Ni/Fe)OOH was also synthesized by the same pathway within a few minutes, which also can prove that the strong oxidation of Fe(NO3)3 is the driving force. We have emphasized this driving force in the revised manuscript from line 143 to 146.

Comment 2: Is it possible to extend this spontaneous synthesis of (Ni/Fe/Mo)OOH electrocatalysts to grow other metal chalcogenides on NF, if it is not a selective synthesis approach.

Response: Thanks for your insightful ideas. Previous studies (Energy Environ. Sci., 2020,13, 3439-3446) have successfully produced the S-(Ni,Fe)OOH electrode, which uses a similar preparation route to dope sulfur atoms.

Comment 3: What is the active metal species in (Ni/Fe/Mo)OOH electrocatalysts for OER and its oxidation state before and after the OER study.

Response: Thank you for your helpful comments. The active metal species in (Ni/Fe/Mo)OOH electrocatalysts are Fe(NO3)3, Ni(OH)2 and MoO3.

The oxidation state before and after the OER study is shown in Figure 3 and S16. And the changes in the oxidation state before and after the OER study is answered using XPS analysis in Comment 4.

Comment 4: The changes in the oxidation state before and after the OER study will be much helpful using XPS analysis.

Response: Thank you for your insightful comments. We have added changes in the oxidation state before and after the OER using XPS analysis. Figure S16 is XPS spectra for (Ni/Fe/Mo)OOH after 72 h of hydrogen production at a current density of 100 mA·cm-2 in 1 M KOH electrolyte. The composition is the same as that of the original electrodes in Figure 3. The components of the catalyst are still Ni(OH)2, FeOOH and MoO3.

We have added the discussion in the revised manuscript from line 239 to 246.

Figure S16. XPS spectra for (Ni/Fe/Mo)OOH after 72h of hydrogen production at a current density of 100 mA·cm-2 in 1 M KOH electrolyte: (a) Survey, (b) Ni 2p, (c) Fe 2p, (d) Mo 3d, (e) O 1s

Comment 5: XRD peak identification for the Fe and Mo compounds is missing.

Response: Thanks for the comment. The original data of XRD is shown in Figure 3a. We have carried out many sets of repeated experiments and tests, and only the peak of nickel can be detected. In Figures 2c and 2d, only the lattice spacing data corresponding to crystalline nickel can be observed, and the molybdenum and iron elements exist in an amorphous state. Hence, there is no crystalline iron and molybdenum in the XRD pattern.

Comment 6: The amount of active material loaded on the NF during the synthesis should be discussed.

Response: Thank you for your advice. We measured the weight of the samples and found that the prepared electrode weighed less than the untreated nickel foam. The nickel can be dissolved into the solution and the prepared electrode become lighter. The catalyst attached on the surface of NF tightly, and cannot be peeled off completely. Therefore, accurate data cannot be obtained. We just get the TEM mapping data of the falling catalyst shards, and the atomic ratio is 40:35:1. Research papers using the similar synthesis method also have not given the specific loading of active ingredients, probably for the similar reason (Energy Environ. Sci., 2020,13, 3439-3446).

Comment 7: OER results of (Ni/Mo)OOH and  (Ni/Fe)OOH for comparison can be productive if it can be formed.

Response: Thanks for your idea. During the exploration of the experiment, we tried to use the same route to prepare nickel-molybdenum electrodes. However, due to the lack of the oxidation effect from Fe(NO3)3 (Energy Environ. Sci., 2020,13, 3439-3446), the preparation was unsuccessful.

The OER results of (Ni/Fe)OOH for comparison is shown in Figure S13 in our manuscript. The best-performing group of (Ni/Fe)OOH catalysts showed an overpotential of 310 mV at a current density of 100 mA·cm-2. The overpotential of (Ni/Fe/Mo)OOH under the same test conditions was 265 mV, much better than that of the (Ni/Fe)OOH samples.

Reviewer 2 Report

Title: Ultrafast and facile synthesis of (Ni/Fe/Mo)OOH on Ni foam 2 for oxygen evolution reaction in seawater electrolysis

This paper reports efficient and cost-effective method to produce electrode for oxygen evolution reaction (OER). The produced electrode shows good performance in alkaline media. The simple method to use reconstruction phenomenon is interesting issue for alkaline OER. Even though this paper showed enhanced performance, the major revision is required for publication. Some comments about this paper are the following:

1.      The author makes the NiFeOOH electrode to show effects of Mo, However, the characterization for NiFeOOH electode is only SEM images with low magnification. Please add the characterization for NiFeOOH electrode with high magnification of SEM and EDS mapping  and XPS.

2.      The role of Mo and Fe would be main factor of this paper. Therefore, synthesis of MoNiOOH electrode is suggested. The performance of MoNiOOH should compared to FeNiOOH and FeNiMoOOH.

3.      In Figure 5, Rs of each electrode is different. Rs is usually ohmic resistance of electrolyte. So Rs should be same for each electrode.

4.      Why the NiFeOOH have reached a stable state in 1 min when there is no Mo? This reaction is the key of this paper. So please explain it.

5.      SEM, TEM and XPS after OER test are suggested. In alkaline media, catalyst can be changed from initial condition. So. Please added it. Figure S10 is too low magnification to represent change. The analysis should be similar with Figure 2.

6.      Atomic ratio of Ni,Fe, and Mo should be added.

7.      The electrochemical result represented using RHE. Please indicates the pH for 1M KOH +1M NaCl and 1M KOH +Seawater.

No comment for quality of English

Author Response

Dear Editor:

We appreciate the opportunity to modify our paper according to the comments of the reviewer. We would like to submit the revised manuscript, along with our response to each comment made by the reviewer as shown below. All the revisions in the revised manuscript are highlighted.

We look forward to receiving your further comments on our revised manuscript. Thanks a lot.

Yours sincerely,

W. Zhang

Tianjin University

Responses to the reviewers’ comments:

Reviewer #1:

Comment 1: The author makes the NiFeOOH electrode to show effects of Mo. However, the characterization for NiFeOOH electode is only SEM images with low magnification. Please add the characterization for NiFeOOH electrode with high magnification of SEM and EDS mapping and XPS.

Response: Thank you for your comments. We added the characterization for the (Ni/Fe)OOH electrode with high magnification of SEM and EDS mapping and XPS, as shown in Figure S9-S12.

As shown in the SEM image in Figure S9, the surface differences of the (Ni/Fe)OOH electrode with different reaction times were less, and there are more wrinkles but no cracks or clustered catalysts compared to the (Ni/Fe/Mo)OOH surface. Figure S10 shows that the (Ni/Fe)OOH electrode surface is smoother. Figure S11 proves that nickel and iron are uniformly distributed. These experiments have shown that it only takes a short time during the preparation of (Ni/Fe)OOH for the surface of the catalyst to be covered with nickel skeletons and tend to be stable. Compared with the three-dimensional shape of the clustered (Ni/Fe/Mo)OOH catalyst, the smaller ECSA of (Ni/Fe)OOH will also affect the performance.

From Figure S12, it can be concluded that the composition of (Ni/Fe)OOH is very similar to that of (Ni/Fe/Mo)OOH and includes FeOOH and Ni(OH)2, named as (Ni/Fe)OOH.

Figure S9. SEM images of (Ni/Fe)OOH at different immersing time, (a) 1 min, (b) 3 min, (c) 5 min

Figure S10. HR-SEM image of (Ni/Fe)OOH (5 min).

Figure S11. EDS mapping images of the (Ni/Fe)OOH sample with the immersing time of 5 min

Figure S12. XPS spectra for (Ni/Fe)OOH (5 min): (a) Survey, (b) Ni 2p, (c) Fe 2p, (d) O 1s

Comment 2: The role of Mo and Fe would be main factor of this paper. Therefore, synthesis of MoNiOOH electrode is suggested. The performance of MoNiOOH should compared to FeNiOOH and FeNiMoOOH.

Response: Thank you for your advice. During the exploration of the experiment, we tried to use the same route to prepare nickel-molybdenum electrodes. However, due to the lack of the oxidation effect from Fe(NO3)3 (Energy Environ. Sci., 2020,13, 3439-3446), the preparation was unsuccessful.

Comment 3: In Figure 5, Rs of each electrode is different. Rs is usually ohmic resistance of electrolyte. So Rs should be same for each electrode.

Response: Thank you for pointing out the question. We remeasured the value of Rs, and the new data are shown in Figure 4e.

Figure 4e. Nyquist plots of the (Ni/Fe/Mo)OOH samples

Comment 4: Why the NiFeOOH have reached a stable state in 1 min when there is no Mo? This reaction is the key of this paper. So please explain it.

Response: Thanks for you remarks. The Fe(NO3)3 has a strong oxidation property and can react with nickel quickly ().  Ni2+ and Fe2+ will combine with OH-, and at the same time react with dissolved oxygen to form Ni(OH)2 and FeOOH (Energy Environ. Sci., 2020,13, 3439-3446). Molybdate (Mo7O246-) dissolved in water will also be oxidized by Fe(NO3)3 to produce MoO3 (Adv. Sci. 2020, 7, 1902034). In this study, (Ni/Fe)OOH was also synthesized by a similar pathway within a few minutes, which also proved that the strong oxidation of Fe(NO3)3 was the key to fast synthesis. We have emphasized this point in the revised manuscript from line 143 to 146.

Comment 5: SEM, TEM and XPS after OER test are suggested. In alkaline media, catalyst can be changed from initial condition. So. Please added it. Figure S10 is too low magnification to represent change. The analysis should be similar with Figure 2.

Response: Thank you for your helpful comments. We added the characterization for the (Ni/Fe/Mo)OOH electrode after OER durability test in 1 M KOH electrolyte and alkaline natural seawater electrolyte using SEM and EDS mapping and XPS, as shown in Figure S14-S18.

Figure S10 in the original manuscript is now Figure S17. We revised it.

Figure S14. SEM image of (Ni/Fe/Mo)OOH catalyst after 72h of hydrogen production at a current density of 100 mA·cm-2 in 1 M KOH electrolyte

Figure S15. EDS mapping images of the (Ni/Fe/Mo)OOH sample after 72h of hydrogen production at a current density of 100 mA·cm-2 in 1 M KOH electrolyte

Figure S16. XPS spectra for (Ni/Fe/Mo)OOH after 72h of hydrogen production at a current density of 100 mA·cm-2 in 1 M KOH electrolyte: (a) Survey, (b) Ni 2p, (c) Fe 2p, (d) Mo 3d, (e) O 1s

Figure S17. SEM image of (Ni,Fe,Mo)OOH catalyst after 72h of hydrogen production at a current density of 100 mA·cm-2 in alkaline natural seawater electrolyte

Figure S18. EDS mapping images of the (Ni/Fe/Mo)OOH sample after 72h of hydrogen production at a current density of 100 mA·cm-2 in alkaline natural seawater electrolyte

Comment 6: Atomic ratio of Ni, Fe, and Mo should be added.

Response: Thank you for your advice. We added the atomic ratio of nickel-molybdenum-iron in the revised manuscript, 40:35:1 according to the TEM-EDS result in Figure S4. We have added this in line 121.

Comment 7: The electrochemical result represented using RHE. Please indicates the pH for 1M KOH +1M NaCl and 1M KOH +Seawater.

Response: Thank you for your helpful comments. The pH value is 14 for 1M KOH +1M NaCl and for 1M KOH +Seawater. All of the measured potentials vs. Hg/HgO were converted to the reversible hydrogen elec-trode (RHE) by the Nernst equation: ERHE = EHg/HgO + 0.0591 × pH + 0.098.

Reviewer 3 Report

In this article "Ultrafast and facile synthesis of (Ni/Fe/Mo)OOH on Ni foam 2 for oxygen evolution reaction in seawater electrolysis" Xu et al reporting the solution phase simple method of synthesis of Ni/Fe/Mo OOH on a nickel foam support. The catalyst is completely characterized. The  catalyst is found to be more competitive to the presently available catalyst known to catalyze the oxygen evolution reaction. The catalyst was found to be durable in the electrolysis of sea water. The catalyst preparation method is a simple way to access the OER catalyst.

As more plagiarism is found in several instances, an extensive editing of English is required. 

In line # 229, four Cdl values (3.25 times, 4.98 times, 6.42 times, and 229 8.14 times) are given for three catalysts. Need clarification on this. 

The article needs minor revision on extensive English editing.

The plagiarism of the article is above 50% while checking along with the references. In the main text, resemblance and plagiarism was found in several instances. It is suggested to rewrite the content to avoid plagiarism

Author Response

Dear Editor:

We appreciate the opportunity to modify our paper according to the comments of the reviewer. We would like to submit the revised manuscript, along with our response to each comment made by the reviewer as shown below. All the revisions in the revised manuscript are highlighted.

We look forward to receiving your further comments on our revised manuscript. Thanks a lot.

Yours sincerely,

W. Zhang

Tianjin University

Responses to the reviewers’ comments:

Reviewer #1:

Comment 1: As more plagiarism is found in several instances, an extensive editing of English is required.

The article needs minor revision on extensive English editing.

Response: Thanks for your advice. We have rewritten the paper and used a professional paper to edit the paper, service from AJE (American Journal Experts).

Comment 2: In line # 229, four Cdl values (3.25 times, 4.98 times, 6.42 times, and 229 8.14 times) are given for three catalysts. Need clarification on this.

Response: Thanks for your suggestion. We apologized for the mistake of the redundant data (8.14 times). We deleted this data in the revised manuscript. In our original experiment, we prepared electrodes with a reaction time of 10 min. Because NiFeOOH has no data of 10 min, we deleted the 10 min data of NiFeMoOOH for a complete comparison.

Round 2

Reviewer 2 Report

The authors well response the reviewers comment So The manuscript is acceptable.